# Impacts of in Utero Heat Stress on Carcass and Meat Quality Traits of Market Weight Gilts

**DOI:** 10.3390/ani11030717

**Published:** 2021-03-06

**Authors:** Jacob R. Tuell, Mariah J. Nondorf, Jacob M. Maskal, Jay S. Johnson, Yuan H. Brad Kim

**Affiliations:** 1Meat Science and Muscle Biology Laboratory, Department of Animal Sciences, Purdue University, West Lafayette, IN 47907, USA; tuell@purdue.edu (J.R.T.); mnondorf@purdue.edu (M.J.N.); 2Livestock Behavior Research Unit, USDA-Agricultural Research Service, West Lafayette, IN 47907, USA; jmaskal@purdue.edu (J.M.M.); Jay.Johnson@ars.usda.gov (J.S.J.)

**Keywords:** carcass composition, fatty acid profiling, gestational heat stress, pork quality, postmortem aging, water-holding capacity

## Abstract

**Simple Summary:**

This study evaluated the effects of exposure of the porcine fetus to in utero heat stress (IUHS) during the first half of gestation on carcass and meat quality attributes when market weight was reached. Pigs exposed to IUHS had lower head and heart weights at slaughter compared to the thermoneutral group. Most measures of carcass quality were not impacted by the treatments, but lower loin muscle area was observed for IUHS carcasses. Additionally, the loins from the heat stressed pigs were found to be tougher, regardless of the duration of aging. Accordingly, minimizing heat stress experienced by gestating pigs would be considered an important factor in improving both yield and quality of pork production systems.

**Abstract:**

This study evaluated the impacts of in utero heat stress (IUHS) on the carcass and meat quality traits of offspring when market weight was reached. Twenty-four F1 Landrace × Large White gilts were blocked by body weight and allocated among thermoneutral (IUTN) or IUHS treatments from d 6 to d 59 of gestation. The offspring were raised under identical thermoneutral conditions, and gilts (*n* = 10/treatment) at market weight (117.3 ± 1.7 kg) were harvested. At 24 h postmortem, the loins (*M. longissimus lumborum*) were obtained, and sections were allocated among 1 d and 7 d aging treatments at 2 °C. Carcasses from IUHS pigs had lower head and heart weights (*p* < 0.05), as well as decreased loin muscle area (*p* < 0.05) compared to IUTN pigs. Loins from the IUHS group had a higher shear force value than the IUTN group (*p* < 0.05). Treatments had no other impacts on carcass and meat quality traits (*p* > 0.05), and Western blots suggested increased toughness of IUHS loins would not be attributed to proteolysis. These results suggest minimizing IUHS during the first half of gestation may be beneficial in improving pork yield and quality, though in general the effects of IUHS would be minimal.

## 1. Introduction

Climate change threatens the efficiency and sustainability of pork production systems [1]. Postnatal heat stress is well documented to adversely affect carcass and meat quality in swine [2]. Exposure of the fetus to maternal hyperthermia, known as in utero heat stress (IUHS), has also been demonstrated to influence postnatal phenotype and performance traits of pigs, leading to dissimilar carcass composition when market weight is reached [3]. The commonly observed differences in carcass composition of IUHS pigs include increased backfat thickness [4,5] and lipid accretion at the expense of protein [6], decreased area of the loin muscle [4], and lower weight of the head [7,8]. However, the effects of IUHS on carcass and meat quality attributes remain poorly understood.

Differences in performance and carcass composition through IUHS are often attributed to differential energy metabolism, as several studies have observed changes in blood insulin levels and glucose metabolism [4,9,10]. Similarly, offspring exposed to IUHS often exhibit increased blood cortisol [11,12], although decreased blood cortisol was observed by Byrd et al. [9] and Maskal et al. [10] following prolonged stress exposure. The level of blood cortisol has been shown to be related to multiple carcass quality attributes such as a positive correlation to carcass weight and backfat thickness and a negative correlation to lean carcass content [13]. Similarly, metabolic differences can influence the rate of muscle acidification through accumulation of hydrogen ions and buildup of postmortem glycolytic metabolites including lactate and pyruvate, thus influencing water-holding capacity and other quality traits of porcine muscle [14,15,16]. Accelerated anaerobic muscle metabolism in glycolytic muscle types early postmortem is known to contribute to protein denaturation and the pale, soft, and exudative condition of pork, a major quality defect [17,18].

While there is evidence to suggest IUHS can influence the carcass characteristics and energy metabolism of porcine muscle, there is at present little literature in regard to meat quality attributes. However, as carcass traits and glycolytic potential are correlated to several pork quality attributes including lightness (determined by CIELAB color space L* values) and drip loss [19], it stands to reason IUHS may impact meat quality in some capacity. Accordingly, the objective of this study was to elucidate the impacts of IUHS on carcass and meat quality traits of market weight pigs.

## 2. Materials and Methods

### 2.1. Animal and Sample Preparation

Procedures involving gestating gilts were approved by the University of Missouri Animal Care and Use Committee (No. 9340). Procedures involving the transported offspring were approved by the Purdue Animal Care and Use Committee (No. 1806001756). Standards regarding animal care and use were in accordance to the *Guide for the Care and Use of Agricultural Animals in Research and Teaching* [20].

The present study was conducted in parallel with Maskal et al. [10], and detailed treatment information regarding live animals can be found there. Briefly, dams utilized in the study were first parity F1 Landrace × Large White gilts (*n* = 24) sired by a single Duroc line (Choice Genetics USA, Des Moines, IA, USA). Pregnant gilts were blocked for similar body weight and allocated among in utero thermoneutral (IUTN; *n* = 12; 17.5 ± 2.1 °C; 70.2 ± 8.8% relative humidity) or cyclical in utero heat stress (IUHS; *n* = 12; 35.8 ± 0.2 °C day and 28.4 ± 0.2 °C night; 80.9 ± 6.0% relative humidity) environmental chambers. Gilts assigned to IUHS were maintained under IUTN conditions for 5 d following insemination, after which they were acclimated to the treatment conditions from d 6 to d 10 at 31.4 ± 2.9 °C day and 26.3 ± 3.0 °C night temperatures at 61.2 ± 21.2% relative humidity. The IUHS treatment was applied from d 11 to d 59 following insemination, after which all gilts were maintained under IUTN conditions (20.9 ± 2.3 °C; 63.6 ± 15.6% relative humidity) until farrowing. Gestating gilts were limit fed 2.0 kg/d to control maternal weight, and after farrowing sows were provided with water and feed ad libitum. The diets were corn and soybean based, formulated to meet or exceed nutrient requirements in accordance to industry standards [21]. At weaning, piglets (*n* = 233) were transported over a period of 11 h and 40 min. Following transport, piglets were housed in groups in 40 mixed sex pens (*n* = 4/pen), and the remaining 73 piglets were not utilized for the study. Pigs utilized for this study were fed the nutrient dense diet for the first 14 d following transport as described by Maskal et al. [10], and all animals were raised under thermoneutral conditions regardless of IUHS environment.

At harvest, gilts (*n* = 10/treatment) weighing 117.3 ± 1.7 kg were blocked by body weight and transported to a harvest facility to be slaughtered under standard conditions. Only gilts were utilized for this study, as differences in carcass traits and other quality attributes between sexes have been previously reported [22]. Gilts were fasted overnight prior to slaughter, and water was provided ad libitum. Weights of the head, liver and gallbladder, heart, kidneys, and spleen were recorded. Hot carcass weight (HCW) was determined as the carcass weight prior to chilling minus the head and viscera. Carcasses were chilled in a 2 °C blast chiller for a 24 h period prior to carcass evaluation and sample collection. Muscle samples for a separate study were collected during carcass chilling, thus chilled carcass weight (CCW) reflects the intact left side of the carcass only. At 24 h postmortem, left sides were ribbed by cutting transversely at the 10th rib and allowed to oxygenate for a 30 min period prior to evaluation as later described. Loins (*M. longissimus lumborum*) were excised from the right side of each carcass and trimmed to a consistent fat depth of approximately 0.6 cm. Loins were transversely sectioned in half and randomly allocated among 1 d postmortem (no further aging) and 7 d postmortem aging treatments at 2 °C under vacuum seal. The sample collections at 1 d served as a baseline information for biochemical and meat quality attributes prior to the occurrence of protein degradation and subsequent meat quality changes at 7 d postmortem aging. Chops (2.54 cm thick) were made from sections to measure meat quality and oxidative stability. One chop from the 1 d postmortem group was overwrap packaged in polyvinylchloride film (23,000 cm^3^ O^2^/m^2^/24 h at 23 °C) for 5 d of aerobic display storage under fluorescent lighting (1800 lx; color temperature = 3500 K) at 2 °C. Following aging or display treatments, samples were frozen/stored at −80 °C until analysis. Biochemical measurements utilized pulverized muscle tissue created by snap freezing lean muscle tissue in liquid nitrogen and powdering with a commercial blender (Waring Commercial, Stamford, CT, USA).

### 2.2. Carcass Evaluation

Following ribbing and the 30 min oxygenation period, carcasses were assessed by a trained evaluator in accordance to National Pork Board Official Color and Marbling Quality Standards (National Pork Board, Des Moines, IA, USA). Subjective lean color scores were assessed using six scale points with 1 being pale and 6 being dark, and marbling scores were assessed using 10 scale points with 1 being practically devoid of marbling and 10 being heavily marbled. Additional subjective measurements were determined by the same trained evaluator using the following criteria: muscle score (1 to 3 with 1 being inferior muscling and 3 being super muscling; half point acceptable), lean firmness score (1 to 3 with 1 being soft and 3 being firm; half point acceptable), and lean wetness score (1 to 3 with 1 being wet and 3 being dry; half point acceptable). Carcass length was determined as the length from the cranial edge of the first rib to the cranial edge of the aitch bone. Fat depths were recorded at the 10th rib and last rib by measuring perpendicular to the skin approximately ¾ of the way up the loin muscle. Loin muscle area was determined by tracing the area of the loin muscle onto acetate paper for later measurement using grids. Standardized fat free lean was determined in accordance to the method utilizing HCW, 10th rib fat depth, and loin muscle area using the method described by the National Pork Producers Council and the American Meat Science Association [23]. Dressing percentage was determined by dividing HCW by BW and expressing on a percentage basis. Percent lean was calculated by dividing standardized fat free lean by HCW and expressing on a percentage basis.

### 2.3. pH and Water-Holding Ability

The pH of the loin samples was determined in duplicate at d 1 and d 7 postmortem by inserting a calibrated meat pH probe (HI 99163, Hanna Instruments, Warner, NH, USA) into the center of the muscle. Prior to measurement, the probe was calibrated with buffers at pH 4 and 7 at 2 °C.

Water-holding ability of loin samples was determined on d 1 and d 7 postmortem samples indirectly through multiple measures including drip loss, purge loss, display weight loss, freezing/thawing loss, and cooking loss. Prior to measurement, samples were gently blotted with paper towels, and all measures were expressed as the percent difference between initial and final weights. Drip loss was measured in accordance to Honikel et al. [24] with modifications described by Kim et al. [25]. Briefly, an approximately 50 g meat cube was trimmed of any visible connective tissue and fat and was suspended in netting in an airtight container for a 48 h period, in accordance to the published method. Purge loss was determined as the percent weight loss of loin sub-sections prior to and after 7 d postmortem aging at 2 °C. Display weight loss was determined on d 1 aged sampled only by measuring weight loss prior to and after 5 d of aerobic display storage as previously described. Freezing/thawing loss was determined as the weight loss before and after freezing of chops at −80 °C after respective aging treatments. Frozen/thawed chops were utilized for cooking loss measurement. Cooking loss was measured using the method described by Kim et al. [26] with modification described by Kim et al. [27]. Chops were cooked to an endpoint temperature of 71 °C and were allowed to cool at room temperature for 30 min prior to measurement. Samples used for cooking loss measurement were wrapped in foil and stored at 4 °C for 16 h prior to shear force determination.

### 2.4. Instrumental Display Color

Surface color of the loin muscles was determined daily in triplicate over the course of the 5 d aerobic display period. CIE L*, a*, and b* values were recorded using a CR-400 Chroma Meter (Konica Minolta, Chiyoda, Tokyo, Japan) with CIE standard illuminant D65. Chroma and hue angle values were calculated in accordance to American Meat Science Association guidelines [28]. After the aerobic display storage period, the samples were frozen at −80 °C for later determination of oxidative stability.

### 2.5. Shear Force

Shear force measurements were determined on 1 d and 7 d postmortem samples after cooking as previously described. At least six cores (1.4 cm diameter) were obtained by cutting parallel to muscle fiber direction. Cores were cut with a TA-XT Plus Texture Analyser (Stable Micro Systems Ltd., Godalming, Surrey, UK) prepared with a 5 kg load cell and Warner-Bratzler V-shaped blade attachment set at a 2 mm/sec test speed. Shear force was determined at the peak force (kg) required to shear, and replicates were pooled prior to statistical analysis.

### 2.6. Western Blotting

Extraction of muscle proteins and gel sample preparation was performed in accordance to Kim et al. [29]. SDS-PAGE load checks were used to confirm gel samples had a similar protein concentration. Western blots of desmin and troponin T were conducted on d 1 and d 7 postmortem samples. Briefly, each lane was loaded with 40 µg of protein. Gels were prepared with 5% stacking and 15% separating bis-acrylamide (100:1) SDS-PAGE separating gels. The same IUTN sample at 7 d postmortem was used as an internal reference for comparison of band intensity. Electrophoresis was conducted for 3 h at 130 V (Bio-Rad Laboratories, Hercules, CA, USA), and proteins were transferred to nitrocellulose membranes for 90 min at 90 V (TE22, Hoefer Inc., Richmond, CA, USA). Blocking of membranes was performed using 5% (*w*/*v*) nonfat dry milk for 1 h at room temperature. Primary antibody incubation was conducted using 3% (*w*/*v*) nonfat dry milk with 1:20,000 dilution of monoclonal mouse anti-desmin (D1022, Sigma Aldrich, St. Louis, MO, USA) or monoclonal mouse anti-troponin T (T6277, Sigma Aldrich, St. Louis, MO, USA) overnight at 4 °C. Washing of membranes was performed three times in PBS-Tween at room temperature. Afterwards, membranes were incubated for 1 h in 3% (*w*/*v*) nonfat dry milk with monoclonal goat anti-mouse IgG (H&L) horseradish peroxidase conjugate (170-6516, Bio-Rad Laboratories, Hercules, CA, USA) at a 1:10,000 or 1:20,000 dilution for desmin and troponin T, respectively. Membranes were washed in the manner previously described. ECL Western blotting reagents (Thermo Fisher Scientific, Waltham, MA, USA) were used to develop membranes, which were subsequently visualized (UVP ChemiDoc-It^TS2^ Imager, Upland, CA, USA). Optical density of the bands was determined in relation to the density of the respective band of the internal reference sample.

### 2.7. Transmission Value

Transmission value was measured in duplicate as a determinant of protein denaturation on 1 d postmortem samples in accordance to the method developed by Ockerman and Cahill [30] with modifications described by Kim et al. [29]. In brief, the sample was mixed with 0.1 M citric acid in 0.2 M sodium phosphate buffer (pH 4.6), after which turbidity was measured using a spectrophotometer (UV-1600PC, VWR International LLC, Radnor, PA, USA) at 600 nm. Higher transmission values indicate a higher extent of protein denaturation.

### 2.8. Lipid Oxidation

Lipid oxidation was determined by assessing the content of malondialdehyde according to the 2-thiobarbituric acid reactive substances (TBARS) assay described by Buege and Aust [31] with the modifications described by Tuell et al. [32]. The assay was conducted in duplicate on the 0 d and 5 d aerobically displayed samples from the 1 d aged group only. The sample filtrate was collected, and absorbance was measured at 531 nm (Epoch, Biotek Instruments Inc., Winooski, VT, USA) and multiplied by 5.54 to determine TBARS value.

### 2.9. Fatty Acid Profiling

Extraction of intramuscular lipids was performed in duplicate from loin samples at 1 d postmortem in accordance to Folch, Lees, and Sloane Stanley [33]. In short, 1.0 g of pulverized muscle tissue was homogenized with 21 mL of 2:1 (*v*/*v*) chloroform to methanol. Sodium methoxide in methanol (0.5 N) was added to prepare fatty acid methyl esters (FAME). Fatty acid profiling was conducted using the conditions described by Tuell et al. [32]. Identification of fatty acids was conducted by comparing retention time to known standards (Supelco 37 components FAME mix, Sigma Aldrich, St. Louis, MO, USA). Fatty acids were expressed as g of fatty acid per 100 g of intramuscular lipid.

### 2.10. Statistical Analysis

The experimental design was a randomized complete block with body weight as the blocking factor. For measures of carcass evaluation, as well as transmission value and fatty acid profile, the IUHS treatment was considered as the fixed effect. Dams and interactions with dam served as random effects. Most measures of meat quality (pH, water-holding ability, instrumental color attributes, shear force, Western blots, and TBARS values) were analyzed in a split-plot design where IUHS treatment served as whole plot and aging or display day as sub-plot. The display color data were analyzed as a repeated measure. The PROC MIXED procedure of SAS (9.4, SAS Institute, Inc., Cary, NC, USA) was used to analyze the data, and means were separated by least significant differences at (*p* < 0.05).

## 3. Results

### 3.1. Carcass Evaluation

As gilts utilized in the study were selected based on similar body weight, no difference in body weight was observed as expected (Table 1; *p* > 0.05). Accordingly, no differences in HCW, CCW, nor dressing percentage were observed (*p* > 0.05). However, gilts from the IUHS group had lower head and heart weights compared to IUTN (*p* < 0.05). No differences in other organ weights including liver and gallbladder, kidney, and spleen were found (*p* > 0.05).

Most measures of carcass quality including lean firmness, wetness, and color, as well as muscle and marbling scores, were unaffected by the IUHS treatment (Table 2; *p* > 0.05). Additionally, the fat depths measured at the 10th rib and last rib were not affected (*p* > 0.05). However, gilts from the IUHS treatment had lower loin muscle area compared to the IUTN group (*p* < 0.05). The mean loin muscle area in the IUHS group was 47.4 square centimeters, 10.4% lower than the mean of 52.9 square centimeters in IUTN. However, likely owing to similar HCW and 10th rib fat depth among treatments, the standardized fat free lean and percent lean values were not affected by the treatments (*p* > 0.05).

### 3.2. Meat Quality

#### 3.2.1. pH and Water-Holding Ability

The pH of the loin muscles was not affected by the IUHS treatment compared to the IUTN control (Table 3; *p* > 0.05), nor was the two-way interaction between IUHS treatment and aging period significant. However, the pH was 0.04 units higher in the 7 d aged group compared to 1 d (*p* < 0.05). No impact of IUHS treatment was observed for any measure of water-holding ability (*p* > 0.05). Improved water-holding was found for 7 d aged samples indicated by lower drip, freezing/thawing, and cooking losses (*p* < 0.05). No two-way interactions were observed in water-holding (*p* > 0.05).

#### 3.2.2. Instrumental Display Color

No measure of display color was affected by IUHS treatment compared to the IUTN controls (Table 4; *p* > 0.05), nor was there any interaction between treatment and display period effects observed (*p* > 0.05). The color attributes were affected, however, by display day (*p* > 0.05), regardless of IUHS treatment. There was a slight increase in CIE L* values over the display period, increasing from 53.7 on d 0 to 54.7 on d 5. Lower CIE a* and CIE b* values were found on d 0 compared to later display days, contributing to lower chroma (color intensity values) for d 0 as well. Hue angle fluctuated throughout the display period, with lower values observed on d 2 and d 5 and higher values found on d 3 and d 4, while d 0 and d 1 were intermediate.

#### 3.2.3. Shear Force and Western Blotting

The main effects of IUHS treatment and postmortem aging duration were significant for shear force values (Figure 1; *p* < 0.05), although the interaction was not significant. The shear force of loin muscles in the IUHS treatment was 2.93 kg, 9.0% higher than the IUTN group (2.69 kg). Loins from the 1 d postmortem group (3.42 kg) had shear force values 35.7% higher than those aged 7 d (2.20 kg).

In accordance to the shear force values, multiple main effects of postmortem aging duration were found for relative abundance of intact and degraded desmin, as well as degraded troponin T (Table 5; *p* < 0.05). Intact desmin (band 1) disappeared with further aging, leading to increased relative abundance of bands 2 and 3. No effect of aging was found in intact troponin T (band 1; *p* > 0.05), although further aging for 7 d increased relative abundance of bands 2 and 3 (*p* < 0.05). No effect of IUHS treatment nor the interaction was observed for relative abundance of desmin nor troponin T (*p* > 0.05). Representative blots for desmin (a) and troponin T (b) are provided in Figure 2.

#### 3.2.4. Transmission Value, Lipid Oxidation and Fatty Acid Profiling

IUHS treatment did not affect lipid oxidation shown by TBARS values (Table 6; *p* > 0.05). However, lipid oxidation increased with aerobic display (*p* < 0.05), regardless of IUHS treatment. Similarly, no effect of IUHS treatment was observed in transmission value, an indication of protein denaturation (*p* > 0.05).

There was no measurable impact of IUHS treatment on the fatty acid profile of intramuscular lipids of pork loin muscle (Table 7; *p* > 0.05).

## 4. Discussion

The present study found several impacts of IUHS treatment on carcass quality of gilts. Specifically, the IUHS treatment reduced weight of the head and heart, as well as decreased the area of the loin muscle. Reduced head size (microcephaly) is in agreement to several published studies [6,7,8]. Microcephaly in pigs has been implicated in impaired neurodevelopment which could impact the ability to cope with stress and increase maintenance costs [34]. However, reduced heart weight of IUHS pigs was not observed by Cruzen et al. [8], although heat stress during finishing did decrease weight of the heart. Other measures of organ weights were unaffected by treatments, in agreement with previous literature [6,7,8]. Decreased loin eye area of IUHS pigs exposed during the first half of gestation compared to thermoneutral counterparts was observed by Boddicker et al. [4]. Cruzen et al. [8] found no impact of IUHS treatment on loin eye area of barrows at slaughter, though Johnson et al. [6] found IUHS to favor lipid accretion at the expense of protein. Accordingly, Boddicker et al. [4] observed increased subcutaneous fat thickness in IUHS pigs compared to IUTN controls, in disagreement with the present study. However, Cruzen et al. [8] found IUHS to not affect the depth of subcutaneous fat of barrows, although IUHS favored lower weights and percentages of skin and bone of the carcass. Dressing percentage was increased in barrows heat stressed in utero during the entirety of gestation compared to IUTN counterparts [8]. However, IUHS during the first half of gestation (as was done in the present study) did not significantly affect dressing percentage compared to IUTN [8], in agreement with the present study. IUHS treatment did not impact the amount of lean tissue or percent lean, in agreement with Cruzen et al. [8], which could owe to gilts being blocked for similar body weight or the aforementioned similarity in subcutaneous fat depth among treatments. Overall, the findings of this study provide some evidence that IUHS would result in postnatal consequences that could adversely impact swine production systems.

The eating quality of pork products consists of juiciness, tenderness, and flavor, which are influenced to an extent by the technological properties including water-holding capacity, proximate composition, oxidative stability, and others [35]. To ensure acceptable quality is achieved for fresh pork products, postmortem aging is utilized [27,36,37]. The present study found postmortem aging to improve the water-holding ability of pork loin as evidenced by lower drip, freezing/thawing, and cooking losses. One factor which may contribute to the improvement in water-holding with further aging would be the slight increase in pH, as the relationship between pH and water-holding capacity of meat products has been well characterized [38]. Additionally, the improvement in water-holding could also be explained by previous purge loss during the aging process, leaving less bulk water available to exit the myofibrillar matrix, or by the fragmentation of myofibrillar protein creating a “sponge effect” as described by Farouk et al. [39]. Previously, Ma et al. [40] found 7 d of further aging to decrease thawing loss of pork compared to 1 d postmortem with no effect on drip and cooking loss, and a slight increase in display weight loss. This may be related to an opposite relationship in pH with extended aging duration observed in the aforementioned study [40]. Several studies have found the pH of fresh meat to increase with aging duration through the generation of basic amino acids during postmortem proteolysis [41,42]. Regardless, the water-holding ability of pork loins was not impacted by IUHS environment. Similarly, no impact of IUHS treatment was found for instrumental display color attributes, although a display period effect was observed. In general, lower CIE a*, CIE b*, and chroma values were observed on d 0 of display, which could be attributed to incomplete oxygenation at the time the d 0 measurement was taken [43]. CIE L* values of pork are less affected by oxygenation time [44] and increased steadily throughout the display period. This may be attributed to increased light scattering as moisture migrates to the meat surface [45]. Numerous factors are known to contribute to the color of fresh meat products including chronic and acute stress pre-slaughter, glycolytic potential, genetics, among others [46]. As previously mentioned, IUHS has been associated with energy metabolism [4,9,10] and the stress response [9,11,12] in swine. However, it appeared such differences did not have any meaningful effect on oxidative stability shown by instrumental display color and TBARS values, nor on the extent of protein denaturation shown by transmission values. This could be partially explained by equivalent fatty acid profile among treatments given the interactive relationship between myoglobin and lipid oxidation [47], as well as the role of lipid profile in influencing oxidative stability [48].

While water-holding ability, instrumental display color attributes, and oxidative stability were not impacted by IUHS treatment compared to IUTN controls, the instrumental tenderness as shown by shear force values was affected. As expected, further postmortem aging improved tenderness in relation to the 1 d postmortem group, in agreement with previous studies [40,42,49]. However, regardless of postmortem aging time, loins from the IUHS group were tougher than those from IUTN controls. Western blot results of desmin and troponin T showed no difference in the degradation of myofibrillar protein with aging between IUHS and IUTN groups. The tenderness of fresh meat products is known to be affected by numerous attributes, and the extent of postmortem proteolysis cannot explain all of the observed variation in meat tenderness [37,50]. One possible explanation could be related to lower birth weights of pigs exposed to IUHS treatment observed in several studies [51,52]. The formation of primary myotubes occurs early in embryonic development and act as a scaffold for secondary myotubes [53], and early in the gestational period is known to be critical for mammalian development [54]. Accordingly, this could explain the inferior loin eye areas observed in the present study and by Boddicker et al. [4], as well as inferior accretion of carcass protein [6]. Supporting this, Gondret et al. [55] found increased muscle fiber diameter and lower number of myofibers in low birthweight gilts compared to higher birthweight littermates. However, Boddicker et al. [4] found no effect of IUHS treatment on muscle fiber diameter or fiber type distribution of the loin muscle, contesting this hypothesis. While results of desmin and troponin T degradation did not support increased toughness of IUHS loin muscles would owe to differential proteolysis, there is some evidence to suggest intrauterine growth restriction can increase shear force through upregulation of calpastatin [56]. At present, however, the mechanism by which IUHS exposure during the first half of gestation may increase meat toughness remains unclear. Additionally, given glycolytic muscle types are highly responsive to postmortem proteolysis with aging [57,58], the effects of IUHS treatment on other (oxidative) muscle types would warrant further study.

## 5. Conclusions

The results of this study provide evidence of the effects of IUHS exposure during the first half of gestation on carcass and meat quality attributes of the offspring. Exposure to IUHS may decrease weights of the head and heart at market weight. Gilts from the IUHS group had lower area of the loin muscle. However, as gilts were blocked by body weight, body weight and fat depths were similar between treatments, resulting in no differences in percent fat free lean. A more thorough understanding of IUHS effects on yield of other primals would be necessary, but at present the impacts of IUHS treatment on yield appear largely minimal. Most measures of meat quality were unaffected by in utero environment. However, the IUHS group had loins with higher shear force values. The results of this study did not provide evidence that higher shear force of the IUHS loins would be attributed to differences in postmortem proteolysis. As such, further study regarding the impacts of IUHS on tenderness of pork muscles would be needed. Minimizing IUHS exposure during the first half of gestation may be beneficial to improve yield and quality of pork production systems. Further study into the mechanisms by which yield and quality attributes could be impacted by IUHS exposure would be warranted.

## Figures and Tables

**Figure 1 animals-11-00717-f001:**
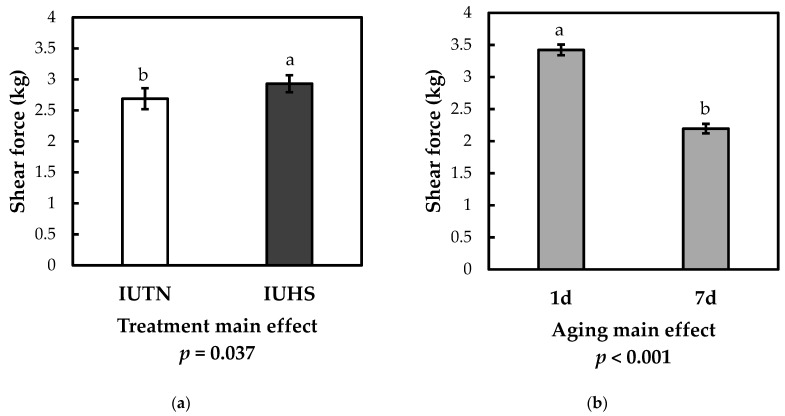
Main effects of in utero heat stress (IUHS) and in utero thermoneutral (IUTN) treatment (**a**) and postmortem aging duration (**b**) on shear force values of pork loin muscle. Different letters a, b indicate least square means different at *p* < 0.05. The two-way interaction was not significant (*p* = 0.160).

**Figure 2 animals-11-00717-f002:**
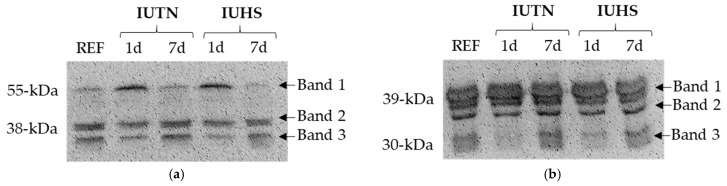
Representative Western blots of desmin (**a**) and troponin T (**b**) from in utero heat stress (IUHS) and in utero thermoneutral (IUTN) treatments at 1 d and 7 d postmortem. REF was same IUTN sample at 7 d postmortem.

**Table 1 animals-11-00717-t001:** Main effect of IUHS treatment on carcass and organ weights of gilts (*n* = 10/treatment).

Trait	Treatment		SEM	*p*-Value
	IUTN	IUHS		
BW ^1^, kg	118.1	116.4	1.7	0.475
HCW ^2^, kg	90.8	88.8	1.4	0.314
CCW ^3^ (left), kg	43.6	42.6	0.7	0.343
Dressing percentage	76.9	76.3	0.4	0.311
Head, kg	5.42 ^a^	5.09 ^b^	0.09	0.021
Liver + Gallbladder, kg	1.49	1.54	0.05	0.553
Heart, kg	0.41 ^a^	0.37 ^b^	0.02	0.046
Kidney, kg	0.31	0.32	0.01	0.995
Spleen, kg	0.17	0.17	0.01	0.563

^a,b^ Least squares means with different letters are different in the same row (*p* < 0.05). ^1^ Body weight. ^2^ Hot carcass weight. ^3^ Chilled carcass weight (left side only).

**Table 2 animals-11-00717-t002:** Main effect of IUHS treatment on carcass attributes of gilts (*n* = 10/treatment).

Trait	Treatment		SEM	*p*-Value
	IUTN	IUHS		
Muscle score (1–3)	2.6	2.4	0.1	0.258
Lean firmness score (1–3)	1.9	1.9	0.2	0.921
Lean wetness score (1–3)	1.9	2.0	0.1	0.382
Lean color score (1–6)	3.0	2.7	0.2	0.328
Marbling score (1–10)	1.4	1.3	0.1	0.517
Carcass length, cm	84.1	84.2	0.7	0.890
Last rib fat, cm	2.6	2.6	0.2	0.718
Fat depth, cm	1.8	1.8	0.1	0.838
Loin muscle area, cm^2^	52.9 ^a^	47.4 ^b^	1.1	0.002
Standardized fat free lean, kg	50.2	48.0	1.0	0.154
Percent lean	55.1	54.0	0.6	0.254

^a,b^ Least squares means with different letters are different in the same row (*p* < 0.05).

**Table 3 animals-11-00717-t003:** Main and interactive effects of IUHS treatment and postmortem aging duration on pH and water-holding ability of pork loin muscle (*n* = 10/treatment).

	Aging Period (Days)	pH	Water-Holding Ability (%)				
			Purge Loss	Drip Loss	Display Weight Loss	Freezing/Thawing Loss	Cooking Loss
*Treatment effect (T)*							
IUTN	-	5.57	6.0	6.0	6.1	5.3	23.0
IUHS	-	5.57	5.8	6.1	6.4	5.2	24.2
SEM	-	0.01	0.8	0.5	0.6	0.5	1.4
*Aging period effect* (*A*)							
1 d	-	5.55 ^y^	-	7.6 ^x^	-	6.1 ^x^	25.2 ^x^
7 d	-	5.59 ^x^	-	4.5 ^y^	-	4.3 ^y^	22.0 ^y^
SEM	-	0.01	-	0.5	-	0.4	1.1
*Two-way interaction* (*T* × *D*)							
IUTN	1 d	5.54	-	7.6	-	6.1	24.3
	7 d	5.59	-	4.5	-	4.4	21.7
IUHS	1 d	5.55	-	7.6	-	6.2	26.1
	7 d	5.59	-	4.6	-	4.2	22.3
SEM		0.01	-	0.6	-	0.6	1.6
*Significance of p-Value*							
Treatment effect (*T*)		0.754	0.905	0.850	0.743	0.938	0.549
Aging period effect (*A*)		<0.001	-	<0.001	-	<0.001	0.012
Two-way interaction (*T* × *A*)		0.688	-	0.942	-	0.714	0.596

^x,y^ Least squares means with different letters are different in the same column (aging period effect; *p* < 0.05).

**Table 4 animals-11-00717-t004:** Main and interactive effects of IUHS treatment and aerobic display storage on instrumental color attributes of pork loin muscle (*n* = 10/treatment).

	Display Period (Days)	CIE L* (Lightness)	CIE a* (Redness)	CIE b* (Yellowness)	Chroma (Color Intensity)	Hue Angle (Discoloration)
*Treatment effect (T)*						
IUTN	-	53.9	5.8	4.7	7.3	36.3
IUHS	-	54.8	6.2	4.3	7.8	36.6
SEM	-	0.8	0.3	0.4	0.5	1.3
*Display period effect (D)*						
0 d	-	53.7 ^c^	5.6 ^d^	4.2 ^bc^	7.0 ^c^	36.3 ^b^
1 d	-	54.4 ^b^	6.3 ^a^	4.8 ^a^	7.9 ^a^	36.8 ^b^
2 d	-	54.1 ^b^	6.3 ^a^	4.3 ^b^	7.7 ^b^	34.1 ^c^
3 d	-	54.4 ^b^	6.0 ^b^	4.8 ^a^	7.7 ^ab^	38.4 ^a^
4 d	-	54.8 ^a^	5.9 ^bc^	4.8 ^a^	7.6 ^b^	38.5 ^a^
5 d	-	54.7 ^a^	5.8 ^cd^	4.1 ^c^	7.1 ^c^	34.8 ^c^
SEM	-	0.6	0.2	0.3	0.3	1.0
*Two-way interaction (T* × *D)*						
IUTN	0 d	53.2	5.5	4.0	6.8	36.1
	1 d	54.1	6.2	4.7	7.8	37.0
	2 d	53.8	6.1	4.2	7.5	34.6
	3 d	53.8	5.8	4.5	7.4	38.0
	4 d	54.1	5.7	4.5	7.3	38.0
	5 d	54.2	5.5	3.8	6.7	34.3
IUHS	0 d	54.3	5.8	4.4	7.3	36.4
	1 d	54.6	6.4	4.8	8.1	36.7
	2 d	54.5	6.5	4.4	7.9	33.7
	3 d	54.9	6.2	5.0	8.0	38.7
	4 d	55.5	6.2	5.0	8.0	39.0
	5 d	55.2	6.0	4.3	7.5	35.2
SEM		0.8	0.3	0.4	0.5	1.5
*Significance of p-Value*						
Treatment effect (*T*)		0.415	0.413	0.488	0.422	0.881
Display period effect (*D*)		<0.001	<0.001	<0.001	<0.001	<0.001
Two-way interaction (*T* × *D*)		0.232	0.791	0.177	0.299	0.727

^a–d^ Least squares means with different letters are different in the same column (display period effect; *p* < 0.05).

**Table 5 animals-11-00717-t005:** Main and interactive effects of IUHS treatment and postmortem aging duration on relative abundance of desmin and troponin T (*n* = 10/treatment).

	Aging Period (Days)	Desmin Band 1	Desmin Band 2	Desmin Band 3	Troponin T Band 1	Troponin T Band 2	Troponin T Band 3
*Treatment effect (T)*							
IUTN	-	1.23	0.85	0.71	1.01	0.86	0.92
IUHS	-	1.10	0.88	0.84	0.98	0.88	0.85
SEM	-	0.05	0.04	0.05	0.03	0.02	0.03
*Aging period effect* (*A*)							
1 d	-	1.33 ^x^	0.73 ^y^	0.61 ^y^	0.98	0.77 ^y^	0.76 ^y^
7 d	-	1.00 ^y^	0.99 ^x^	0.94 ^x^	1.01	0.97 ^x^	1.00 ^x^
SEM	-	0.04	0.03	0.04	0.03	0.02	0.02
*Two-way interaction* (*T* × *A)*							
IUTN	1 d	1.39	0.72	0.58	0.98	0.75	0.73
	7 d	1.06	0.97	0.84	1.03	0.96	0.96
IUHS	1 d	1.27	0.74	0.64	0.98	0.79	0.79
	7 d	0.93	1.02	1.04	0.99	0.98	1.05
SEM		0.05	0.05	0.06	0.04	0.03	0.04
*Significance of p-Value*							
Treatment effect (*T*)		0.073	0.581	0.067	0.655	0.412	0.111
Aging period effect (A)		<0.001	<0.001	<0.001	0.132	<0.001	<0.001
Two-way interaction (*T* × *A*)		0.901	0.655	0.130	0.347	0.740	0.427

^x,y^ Least squares means with different letters are different in the same column (aging period effect; *p* < 0.05).

**Table 6 animals-11-00717-t006:** Main and interactive effects of IUHS treatment and aerobic display storage on lipid oxidation and transmission value of pork loin muscle (*n* = 10/treatment).

	Display Period (Days)	TBARS Value ^1^	Transmission Value (%)
*Treatment effect (T)*			
IUTN	-	0.64	46.0
IUHS	-	0.64	46.1
SEM	-	0.03	3.4
*Display period effect (D)*			
0 d	-	0.51 ^b^	-
5 d	-	0.76 ^a^	-
SEM	-	0.02	-
*Two-way interaction (T* × *A)*			
IUTN	0 d	0.53	-
	5 d	0.75	-
IUHS	0 d	0.50	-
	5 d	0.77	-
SEM		0.03	-
*Significance of p-Value*			
Treatment effect (*T*)		0.957	0.982
Display period effect (D)		<0.001	-
Two-way interaction (*T* × *D*)		0.070	-

^a,b^ Least squares means with different letters are different in the same column (display period effect; *p* < 0.05). ^1^ TBARS value expressed as mg malondialdehyde per kg meat.

**Table 7 animals-11-00717-t007:** Main effect of IUHS treatment on fatty acid profile of pork loin muscle (*n* = 10/treatment).

	Treatment			
Fatty Acid ^1^	IUTN	IUHS	SEM	*p*-Value
C4:0	N.D. ^2^	N.D.	-	-
C6:0	N.D.	N.D.	-	-
C8:0	N.D.	N.D.	-	-
C10:0	0.13	0.12	0.01	0.195
C12:0	0.03	0.03	<0.01	0.494
C14:0	1.30	1.27	0.04	0.673
C14:1	0.23	0.19	0.05	0.619
C16:0	24.01	24.08	0.24	0.825
C16:1	3.48	3.26	0.12	0.192
C18:0	10.66	11.05	0.22	0.233
C18:1n-9*c*	37.21	37.27	0.61	0.953
C18:1n-9*t*	0.32	0.33	0.01	0.347
C18:2n-6*c*	11.66	11.44	0.46	0.742
C18:2n-6*t*	0.07	0.04	0.02	0.361
C18:3n-3	0.27	0.24	0.02	0.172
C18:3n-6	0.07	0.05	0.01	0.128
C20:0	0.15	0.14	0.01	0.748
C20:1n-9	0.46	0.47	0.02	0.689
C20:2	0.27	0.28	0.01	0.513
C20:3n-3	0.01	0.03	0.01	0.249
C20:3n-6	0.26	0.27	0.02	0.746
C20:4n-6	2.17	2.14	0.15	0.900
C20:5n-3	0.01	0.01	<0.01	0.161
C22:0	0.04	0.04	0.01	0.627
C22:1n-9	N.D.	N.D.	-	-
C22:6n-3	0.11	0.01	<0.01	0.230
C24:1n-9	N.D.	N.D.	-	-
Total SFA ^3^	36.55	37.05	0.38	0.367
Total MUFA ^4^	41.88	41.71	0.63	0.848
Total PUFA ^5^	14.83	14.51	0.66	0.736
Total UFA ^6^	56.73	56.23	0.27	0.204

^1^ Expressed as g fatty acid per 100 g intramuscular lipid. ^2^ Not detected. ^3^ Total saturated fatty acids. ^4^ Total monounsaturated fatty acids. ^5^ Total polyunsaturated fatty acids. ^6^ Total unsaturated fatty acids.

## Data Availability

Data are available on request to the corresponding author.

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
