# Peer review of "Impacts of in Utero Heat Stress on Carcass and Meat Quality Traits of Market Weight Gilts"

_animals, 2021, doi:10.3390/ani11030717_

Round 1

Reviewer 1 Report

Comments and Suggestions for Authors

The manuscript contributes further knowledge on carcass and meat quality attributes of market weight gilts when exposed to in utero heat stress (IUHS). The results of this study provide a basis of knowledge for further studies on the effects of IUHS on meat quality attributes. The objectives of this manuscript are of interest and fit well within the scope of this journal.

The work appears sufficiently well conducted and the results reported are important due to the pressures of climate change. This is a well written manuscript and it is evident that the authors have great knowledge in this research area.

Followed is some suggestions for improvement:

L61 When mentioning CIE L* it would be important to mention color space so the reader understands when you write ‘lightness’ that you are referring to color.

L75 please indicate that gilts were blocked for ‘similar’ bodyweight in the materials and methods and insert the avg. weight.

L93 Please state why only gilts were chosen for harvesting.

L124 Place a gap after ‘1’

L149 State why 48hrs was the time chosen.

L237 add in ‘similar’ before bodyweight.

L278 add in ‘d’ before 5 to be consistent with next sentence.

Author Response

Authors’ response:

The authors appreciate the editor’s and reviewers’ comments and for the assistance in preparing a more accurate manuscript. Specific steps taken to implement the suggestions of the reviewers are detailed below (Lines refer to lines in the original, submitted version of the manuscript and coincide with specific lines of concern highlighted by the reviewer).

Editor:

lines 25-26 - Biologically, it is obvious that tenderness increases as the tissue ages, please remove this sentence.

Authors’ response:

The sentence describing the decrease in shear force through postmortem aging has been deleted and abbreviated as follows:

Loins from the IUHS group had a higher shear force value than the IUTN group (p < 0.05).

lines 227-234- Please describe the statistical model used for each analysis since the split pot was a factorial. Indicate each individual effect and their specific levels.

Authors’ response:

The statistical model description has been further described as follows:

For measures of carcass evaluation, as well as transmission value and fatty acid profile, the IUHS treatment was considered as the fixed effect. Dams and interactions with dam served as random effects. Most measures of meat quality (pH, water-holding ability, instrumental color attributes, shear force, western blots, and TBARS values) were analyzed in a split-plot design where IUHS treatment served as whole plot and aging or display day as sub-plot. The display color data were analyzed as a repeated measure.

324-344 Please discuss the implications of the effects on this quantitative attributes (weight of head, heart, and loin size. Why those differences are important for the industry and consumer?

Authors’ response:

The following sentence has been added to explain the implications of reduce head size:

Microcephaly in pigs has been implicated in impaired neurodevelopment which could impact the ability to cope with stress and increase maintenance costs [33].

The following sentence has been added to explain the implications of reduce loin size to industry:

Overall, the findings of this study provide some evidence that IUHS would result in postnatal consequences that could adversely impact swine production systems.

386-388 Why in this experiment the aging interval was from 1 – 7 days. Why not 7 and 14 for example. It is known that fresh meats are indeed transported for at least 4-10

Authors’ response:

The authors appreciate the request for clarification by the editor. We agree that utilizing 7 and 14 d would be more appropriate to industry practices. However, we believe that in order to assess if proteolysis and tenderization would be impacted by measuring WBSF of chops at 1 and 7 days was most appropriate. Collecting biochemical and quality data from the early postmortem samples (e.g. 1 day) would provide a baseline information prior to the occurrence of considerable protein degradation and subsequent meat quality changes. It has been demonstrated that significant tenderization of pork loin muscle occurs from 2 to 9 days postmortem, while WBSF values between 9 and 16 day aged groups are comparable (see https://onlinelibrary.wiley.com/doi/epdf/10.1111/j.1745-4573.1998.tb00661.x). Accordingly, it could be expected that no differences among treatment groups would be observed from 7 to 14 days.

Reviewer 1:

The manuscript contributes further knowledge on carcass and meat quality attributes of market weight gilts when exposed to in utero heat stress (IUHS). The results of this study provide a basis of knowledge for further studies on the effects of IUHS on meat quality attributes. The objectives of this manuscript are of interest and fit well within the scope of this journal.

The work appears sufficiently well conducted and the results reported are important due to the pressures of climate change. This is a well written manuscript and it is evident that the authors have great knowledge in this research area.

Followed is some suggestions for improvement:

L61 When mentioning CIE L* it would be important to mention color space so the reader understands when you write ‘lightness’ that you are referring to color.

Authors’ response:

The requested change has been made:

However, as carcass traits and glycolytic potential are correlated to several pork quality attributes including lightness (determined by CIELAB color space L* values) and drip loss [19].

L75 please indicate that gilts were blocked for ‘similar’ bodyweight in the materials and methods and insert the avg. weight.

Authors’ response:

The sentence on L175 has been modified to include that pregnant gilts were blocked by similar body weight. However, body weight of the gilts is not available or published in the parallel study by Maskal et al. (2020).

L93 Please state why only gilts were chosen for harvesting.

Authors’ response:

The following sentence has been added beginning on L95:

Only gilts were utilized for this study, as differences in carcass traits and other quality attributes have been previously reported [22].

L124 Place a gap after ‘1’

Authors’ response:

The requested change on L124 has been made.

L149 State why 48hrs was the time chosen.

Authors’ response:

The following clarification has been made beginning on L149:

Briefly, an approximately 50 g meat cube was trimmed of any visible connective tissue and fat and was suspended in netting in an airtight container for a 48 h period, in accordance to the published method.

L237 add in ‘similar’ before bodyweight.

Authors’ response:

The requested change on L237 has been made.

L278 add in ‘d’ before 5 to be consistent with next sentence.

Authors’ response:

The requested change on L278 has been made.

Reviewer 2 Report

The manuscript deals with a theme that is not studied so often. It has merit, it is well conducted and discussed by the authors. Is possible to notice that the authors did a great effort in trying to research a theme such as how heat stress in utero affects meat quality, which still has many gaps to fully understand the mechanism. Even though, the authors did an extensive research through many variables studied, which are well presented.

Author Response

Reviewer 2:

The manuscript deals with a theme that is not studied so often. It has merit, it is well conducted and discussed by the authors. Is possible to notice that the authors did a great effort in trying to research a theme such as how heat stress in utero affects meat quality, which still has many gaps to fully understand the mechanism. Even though, the authors did an extensive research through many variables studied, which are well presented.

Authors’ response:

The authors thank Reviewer 2 for their comments regarding this manuscript.